# Two single-point mutations in Ankyrin Repeat one drastically change the threshold temperature of TRPV1

Shogo Hori[1], Michihiro Tateyama[2], Tsuyoshi Shirai [1], Yoshihiro Kubo [2] & Osamu Saitoh [1,3] ✉

TRPV1 plays an important role in the thermosensory system; however, the mechanism controlling its heat activation property is not well understood. Here, we determine the heat activation properties of TRPV1 cloned from tailed amphibians, which prefer cooler environments, finding the threshold temperatures were approximately 10 °C lower compared with rat TRPV1 (rTRPV1). We find that two amino acid residues (Gln, Leu/Val) in the Ankyrin Repeat 1 (ANK1) region of the N-terminal domain are conserved among tailed amphibians and different from those (Arg, Lys) in rTRPV1. We observe the activation by heat in all urodelan TRPV1s is markedly elevated by substitution of these two amino acids. Conversely, reciprocal substitutions of rTRPV1 apparently lowers the high threshold temperature. Our studies demonstrate that tailed amphibians express TRPV1 with a reduced heat-activation threshold by substitution of two amino acid residues in the ANK1 region that likely contribute to cool-habitat selection.

Various animals live and thrive throughout the planet including frigid polar zones and the sunbaked desert. Thermal perception is an essential sensory system for animal survival and thermosensory, and thermoregulatory systems have developed through evolution and adaptation to various environments. Subsets of transient receptor potential (TRP) channels (thermoTRPs) function as thermal sensors and they are specifically activated over distinct temperature ranges. TRPA1, TRPV1, TRPM8, and TRPC5 are primarily expressed in nociceptive sensory neurons and function as sensors for noxious thermal and chemical stimuli[1–4]. In particular, TRPV1 is a major sensor for noxious heat and is activated above 43 °C in rodents and humans[5]. As to the structural determinant(s) of the thermal activation, the contributions of distinct regions of TRPV1 have been reported as follows. (1) The deletions of C-terminal of TRPV1 lowered the threshold for the thermal activation[6–8]. (2) The membrane-proximal domain (MPD) connecting N-terminal Ankyrin repeats (ANKs) and the first transmembrane domain determines the temperature dependence of channels[9,10]. (3) Heat activation and shifts of threshold temperature is intrinsic in the pore domain of TRPV1[11,12]. (4) The ANKs domain is important for heat activation of TRPV1[13,14]. From these functional studies, it appears that molecular determinants of hypothetical heat-dependent activation may rather spread over the TRPV1 molecule.

More recently, Nadezhdin et al. reported the cryo-EM structures of heat-activated TRPV3[15], and Kwon et al. showed the cryo-EM structures of heat-activated TRPV1[16]. Although detailed information of conformational changes are different, both studies demonstrated that the mechanism of heat activation includes two steps, and that global conformational changes across multiple topologically distant subdomains of TRP channel might be followed by the rearrangement of the pore to lead to gate opening. Therefore, since each distinct domains might be involved after heat stimulation, it is thought that mutations that affect heat-sensing or coupling mechanisms could not be functionally distinguishable. Thus, the primary heat-sensing module(s) which determine the threshold temperature of TRPV1 still remains unclear at this stage.

[1]Graduate School of Biosciences, Nagahama Institute of Bio-Science and Technology, Nagahama Shiga 526-0829, Japan. [2]Division of Biophysics and Neurobiology, National Institute for Physiological Sciences, Okazaki, Aichi 444-8585, Japan. [3]Genome Editing Research Institute, Nagahama Institute of Bio-Science and Technology, Nagahama Shiga 526-0829, Japan. ✉e-mail: o_saito@nagahama-i-bio.ac.jp

Because they are dependent on various habitat temperatures, every animal must have a specific thermal sensing process. TRPV1 plays an important role in the thermosensory and thermoregulatory systems of animals and its heat-activation threshold determines its temperature-sensing properties. Investigation of TRPV1 from ground squirrels and camels, which can tolerate higher environmental temperatures, revealed that these animals express TRPV1 with a dramatically decreased thermosensitivity[13]. Furthermore, characterization of a cold-sensitive thermoTRP, TRPM8 from elephants and penguins indicated that penguin's TRPM8 exhibited much decreased cold-activated currents[17]. In these reports, several residues and structures involved in tuning thermal activation in thermoTRPs have been reported, such as Asn126 and Glu190 of squirrel TRPV1 in the N-terminus, Asn124 and Glu188 of camel TRPV1 in the N-terminus[13], and Y919 of penguin TRPM8 in the pore domain[17]. Here, we focus on TRPV1 channels from members of the order Urodela, which are tailed amphibians. Since most of these animals prefer cooler environments, we hypothesize that these animals express unique TRPV1s exhibiting a lower threshold temperature that contributes to habitat selection. First, we cloned axolotl TRPV1 (axTRPV1) and find that it is heat-activated above 31 °C, which is strikingly lower compared with that of rat TRPV1 (43 °C)[18]. This property of axTRPV1 is consistent with the heat-induced behavior of axolotls. Therefore, to determine whether this lower threshold temperature of TRPV1 is a common feature among tailed amphibians and further understand the underlying molecular mechanism that control heat-activation thresholds of TRPV1 channels, we isolated cDNAs that encode TRPV1 channels from three other tailed amphibians including the Iberian ribbed newt (IR Newt, *Pleurodeles waltl*), Yamato salamander (Y Sal, *Hynobius vandenburghi*), and Jananese clawed salamander (JC Sal, *Onychodactylus japonicus*). The IR Newt is a species native to the Iberian Peninsula and Morocco, and it prefers a temperature range of 15–28 °C[19]. Y Sal is a lowland and still-water breeder distributed throughout the Kinki and Chubu regions of Japan[20,21]. Its breeding behavior in water begins in winter. JC Sal is a stream-breeding

salamander found in the montane regions of Japan[22,23]. The eggs of this salamander are laid at headwater or in underground water. We express the TRPV1 cDNAs of four tailed amphibians and rat in *Xenopus* oocytes and conducted an electrophysiological analysis. The results indicate the TRPV1 channels from all four tailed amphibians are heat-activated at lower temperatures between 28 °C and 34 °C. By characterizing chimera and point mutants of the TRPV1s from axolotls and rats, we demonstrate that two amino acid residues in the Ankyrin Repeat 1 (ANK1) region of the N-terminal domain are required to determine the heat-activation threshold of TRPV1. These are Q128 and L154 in axTRPV1 and R114 and K140 at the corresponding positions in rat TRPV1. We also find that two amino acids at the same positions of other urodelan TRPV1s play an important role in reducing their heat-activation threshold. Further, we examine the molecular mechanism controlling the heat-activation threshold of TRPV1 by homology structural modeling and by analyzing the thermodynamic properties of the amino acids at the two key positions. Finally, biochemical analysis indicates that the protein structure stability of the N-terminus is efficiently controlled by these two amino acids of ANK1, providing a simple mechanism determining the heat-activation threshold of TRPV1 in amphibians.

## Results

### Behavioral response to heat of tailed amphibians

We previously examined noxious effects of mild heat (33 °C) on axolotl behavior and compared these effects with that of the IR Newt. We observed that axolotl movements were induced significantly by stimulation with mild heat from 33 °C, whereas those of the IR Newt were induced at 39 °C[18]. We further examined the behavioral responses to heat of Y Sal and JC Sal. The results indicated that Y. Sal exhibits a noxious response at 36 °C, but the noxious behavior of JC Sal is detected from 20 °C (Fig. 1). Therefore, these animals show locomotion to avoid heat at a lower threshold temperature, which is distinctly lower than the threshold of mammalian TRPV1 (43 °C)[5].

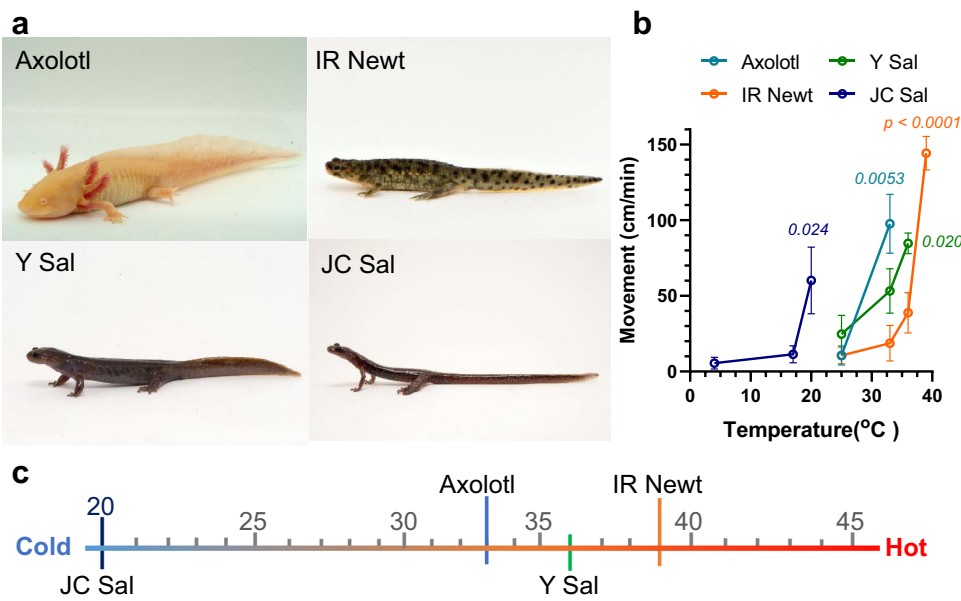

Fig. 1 | **Behavioral responses to heat in tailed amphibians. a** Axolotls (ax, *Ambystoma mexicanum*), Iberian ribbed newt (IR Newt, *Pleurodeles waltl*), Yamato salamander (Y Sal, *Hynobius vandenburghi*), and Japanese clawed salamander (JC Sal, *Onychodactylus japonicus*). **b** Noxious heat behavior of tailed amphibians. Each animal was placed into a metallic tray pre-warmed at various temperatures. The locomotor activity was monitored using a video camera and the mean movement was calculated. Each data point represents the mean ± SE. Statistical significance for the differences with the control group was determined by one-way ANOVA followed by Dunnett's multiple comparison test, and *P* values given in italics indicate statistical significance when <0.05. (*n* = 3 for IR Newt and Y Sal, 4 for ax, 5 for JC Sal). Source data are provided as a Source Data file. **c** The temperature at which a significant increased locomotion of each urodelan was detected was summarized.

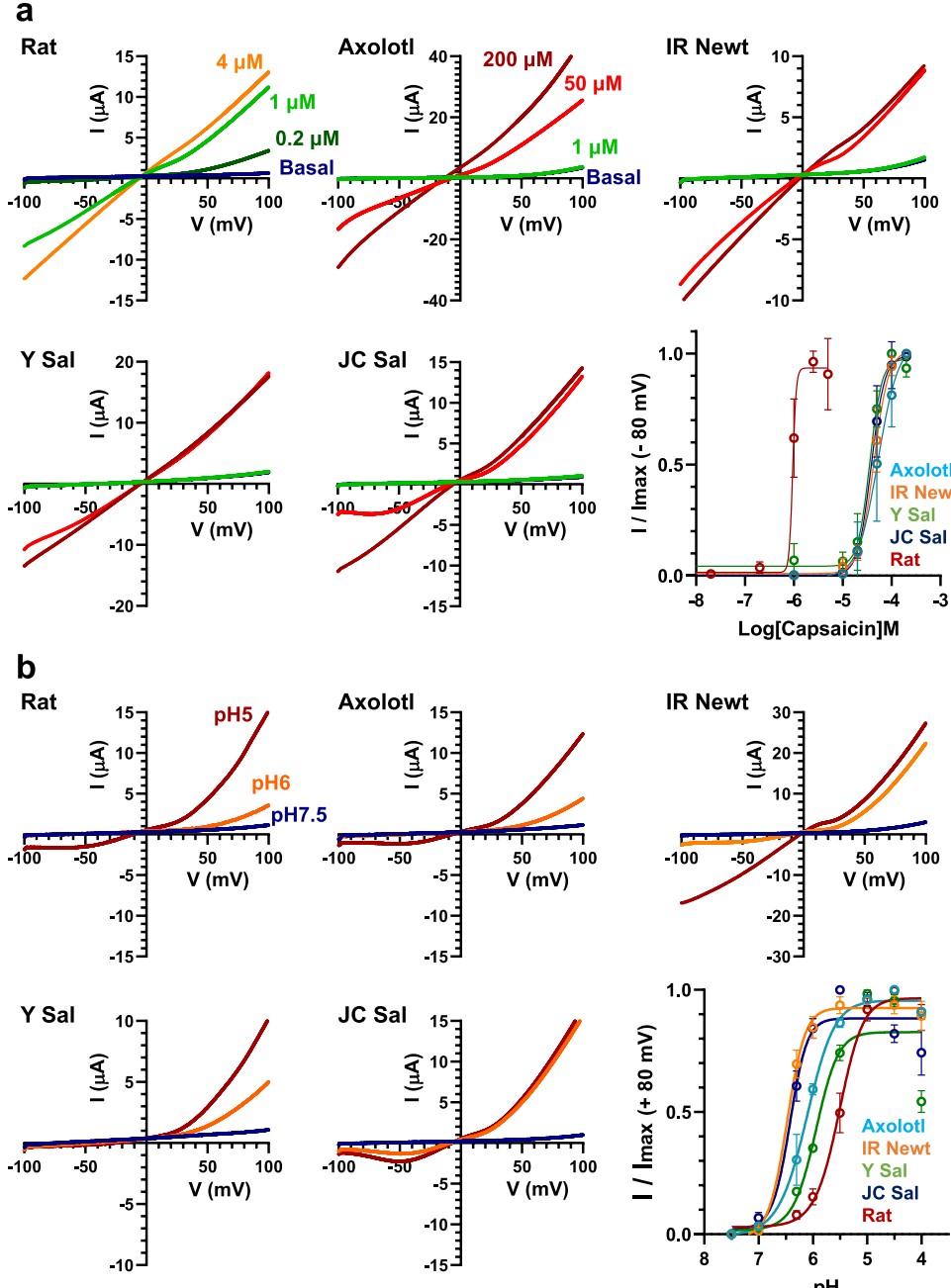

**Fig. 2 | Chemical sensitivities of TRPV1s from tailed amphibians. a** Response to capsaicin of TRPV1s from four tailed amphibians. Each TRPV1 was expressed in *Xenopus* oocytes and the capsaicin-induced response was examined by the two-electrode voltage clamp method. Voltage ramps from −100 mV to +100 mV were applied every 2 s and capsaicin was added at various concentrations. The *I–V* relationship for the max response at the indicated concentration of capsaicin is shown. The dose-response to capsaicin for each TRPV1 cDNA at −80 mV is summarized (*n* = 4 for each concentration of capsaicin). Each data point represents the mean ± SE. Source data are provided as a Source Data file. **b** Response to acid of TRPV1s from four tailed amphibians. Each TRPV1 was expressed in *Xenopus* oocytes and the acid-induced response was examined by the two-electrode voltage clamp method. Voltage ramps from −100 mV to +100 mV were applied every 2 s and acid at various concentrations were added. The *I–V* relationship for the max response at the indicated concentration of acid is shown. The dose–response to acid for each TRPV1 at + 80 mV is summarized (*n* = 4 for each pH). Each data point represents the mean ± SE. Source data are provided as a Source Data file.

## TRPV1s from tailed amphibians exhibit a lower threshold temperature for heat-activation

In addition to TRPV1 from axolotls (axTRPV1)[18], we isolated cDNAs encoding TRPV1 from the IR Newt, Y Sal, and JC Sal. Phylogenetic analysis using the deduced amino acid sequence of the TRPV1s from tailed amphibians revealed one clade, which is diverse from the clade containing frog TRPV1 (Supplementary Fig. 1). We next examined their chemical sensitivities using a two-electrode voltage clamp analysis in *Xenopus* oocytes expressing each of the four tailed amphibian TRPV1s.

Capsaicin and acid activated all TRPV1s in a dose-dependent manner (Fig. 2). These results suggest that TRPV1s from tailed amphibians share similar chemical sensitivities with mammalian TRPV1s to some extent.

We next examined the thermal-activation property of TRPV1s from the tailed amphibians and compared with that of the rat (rTRPV1). Heat stimulation up to 40 °C did not significantly activate rat TRPV1, but activated all of the urodelan TRPV1s. When their mean currents were induced at 40 °C and compared with rat TRPV1, a significantly

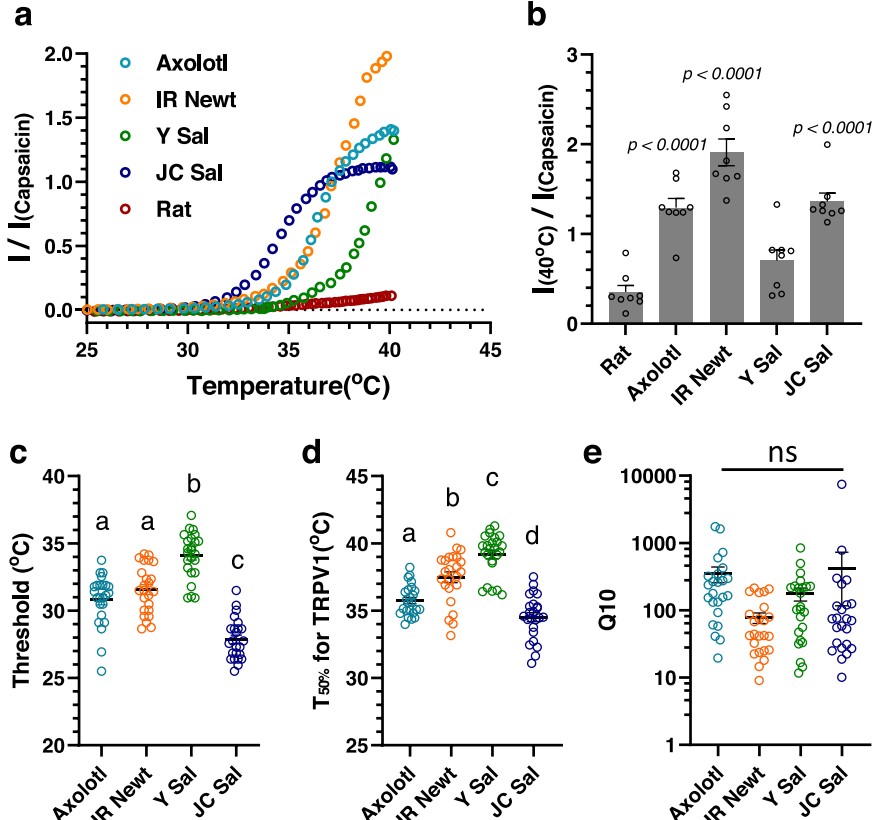

**Fig. 3 | Thermal-activation property of TRPV1s from tailed amphibians.**
**a** Response to heat of the TRPV1s from four tailed amphibians. Each TRPV1 was expressed in *Xenopus* oocytes and the heat-induced response was examined by the two-electrode voltage clamp method as described in the "Methods" section. The representative heat-induced currents standardized by the capsaicin-induced currents (5 μM for rat and 50 μM for urodelan TRPV1s) of each TRPV1 are indicated. **b** Average currents for heat stimulation at 40 °C standardized by the capsaicin-induced currents of each tailed amphibian TRPV1 were compared ($n = 8$). Each data point represents the mean ± SE. Statistical significance for the difference with rat TRPV1 was determined by one-way ANOVA followed by Dunnett's multiple comparison test, and *P* values given in italics indicate statistical significance when <0.05. Source data are provided as a Source Data file. **c** Average threshold temperatures of each tailed amphibian TRPV1 were compared ($n = 24$). The means ± SE

was indicated. Statistically significant differences were determined by one-way ANOVA with Tukey–Kramer multiple comparison method. Different letters (**a**, **b** or **c**) indicate significant differences between the four groups ($p < 0.0001$). Source data are provided as a Source Data file. **d** Average half-maximal activation temperatures of each tailed amphibian TRPV1 were compared ($n = 24$). The means ± SE was indicated. Statistically significant differences were determined by one-way ANOVA with Tukey–Kramer multiple comparison method and different letters (**a**, **b**, **c**, or **d**) indicate significant differences between the four groups ($p < 0.05$). Source data are provided as a Source Data file. **e** Average $Q_{10}$ values of each tailed amphibian TRPV1 were compared ($n = 24$). The means ± SE was indicated. Statistically significant differences were determined by one-way ANOVA with Tukey–Kramer multiple comparison method, but were not detected (NS). Source data are provided as a Source Data file.

elevated activation was observed for axTRPV1, IR Newt TRPV1, and JC Sal TRPV1 (Fig. 3a, b; Supplementary Fig. 2). Arrhenius plot analysis revealed temperature thresholds for heat-activation of 30.91 ± 0.34 °C, 31.57 ± 0.36 °C, 34.11 ± 0.35 °C, 27.87 ± 0.31 °C for axTRPV1, IR Newt TRPV1, Y Sal TRPV1, and JC Sal TRPV1, respectively (Fig. 3c). These threshold temperatures were quite lower compared with the reported threshold for rTRPV1 of 43 °C[5]. The half-maximal activation temperatures for each TRPV1 were also determined (Fig. 3d). Furthermore, all urodelan TRPV1s exhibited relatively high $Q_{10}$ values (Fig. 3e). The *I–V* relationships of oocytes expressing each TRPV1 at different temperatures (25, 30, 35, and 40 °C) are shown (Supplementary Fig. 3).

**The N-terminal end containing Ankyrin Repeat 1 determines the lower threshold of axolotl TRPV1 and the higher threshold of rat TRPV1**
We took advantage of the chimeric constructs to probe structural determinants of axTRPV1 that are required for lower temperature activation (LTA). We examined the heat-activation threshold of chimeric constructs expressed in *Xenopus* oocytes electrophysiologically (Fig. 4a–d). Because intrinsic currents induced by heat stimulation above 40 °C result from the cell damage observed in cRNA

non-injected control oocytes, heat stimulation was performed up to 40 °C for the oocyte experiments. When the whole C-terminus containing all of the axTRPV1 transmembrane domains was replaced with the corresponding region of rTRPV1, this chimera channel (AR) exhibited LTA (Fig. 4b). Next, we examined the roles of each Ankyrin Repeat (ANK1-6) of axTRPV1 in LTA by constructing five chimeras (see Fig. 4a, ARAC1-5); however, only ARAC1 exhibited LTA (Fig. 4c). The heat-activation threshold of rTRPV1 was markedly lowered by exchanging only the N-terminal end containing ANK1 of axTRPV1. These results suggest that the N-terminal end of 166 amino acids (AA), which contains ANK1, contributes to LTA.

To study whether the corresponding N-terminal end of rTRPV1 of 152 AA is capable to, reversely, elevate the threshold temperature of axTRPV1, we constructed another chimeric channel RAAC1 (see Fig. 4a). The threshold temperature of rTRPV1 is 43 °C[4]. Therefore, to monitor the heat-activation of channels above 40 °C, whole cell patch-clamp recording using HEK293T expression system was performed (Fig. 4d). As demonstrated by oocyte experiments, ARAC1 chimera showed the LTA (the threshold: 28.0 ± 0.93 °C) as axTRPV1 (the threshold: 31.47 ± 1.28 °C). The heat-activation threshold of RAAC1 was determined as 41.56 ± 0.90 °C, which is similar to that of rTRPV1

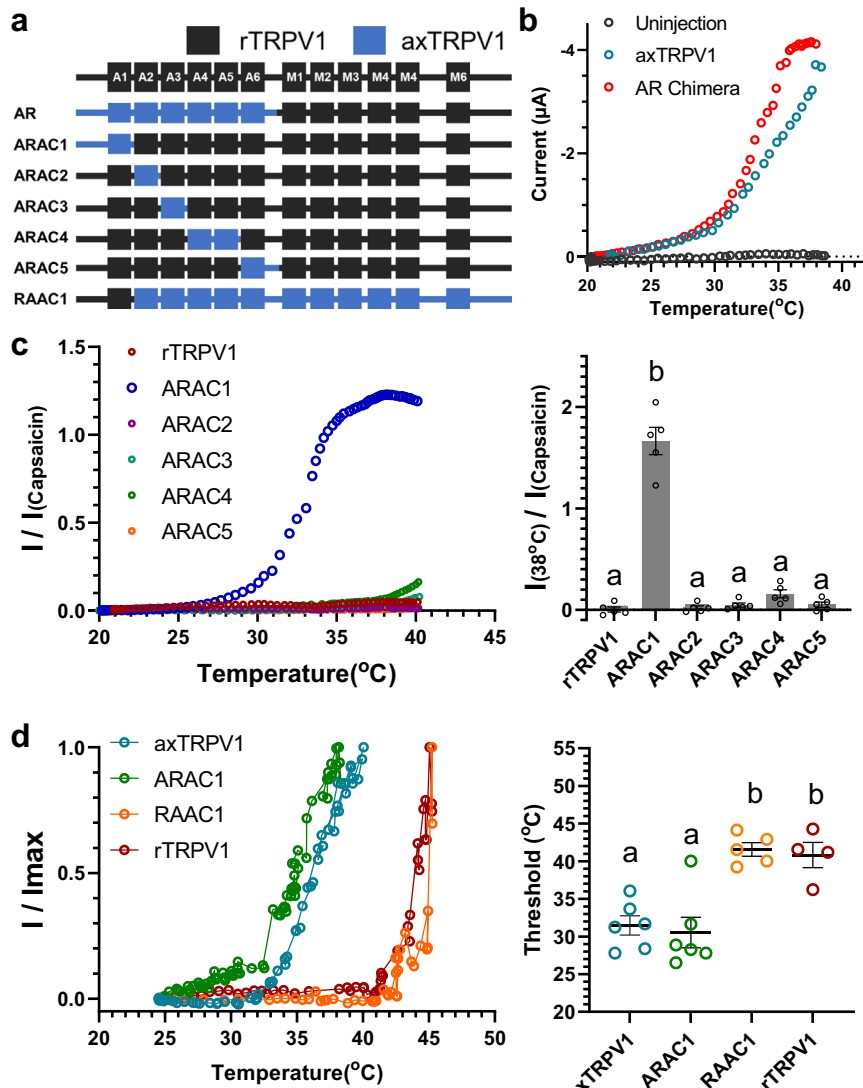

**Fig. 4 | The N-terminal end containing an Ankyrin Repeat 1 determines the heat-activation threshold of axolotl and rat TRPV1. a** Schematic representation of TRPV1 chimeras. Chimeric channels were generated between rat (black) and axolotl TRPV1 (Blue) on the basis of the location of the ankyrin repeats (A1–A6) and transmembrane domains (M1–M6). **b** AR chimera or axTRPV1 was expressed in *Xenopus* oocytes and the representative heat-induced currents were examined by the two-electrode voltage clamp method as indicated. **c** ARAC chimera (ARAC1-5) or rTRPV1 was expressed in *Xenopus* oocytes and the representative heat-induced currents were standardized by the capsaicin-induced currents at 5 μM were indicated. Average current for heat stimulation at 38 °C standardized by the capsaicin-induced currents of each chimera in oocytes were compared (*n* = 5). Each data point represents the mean ± SE. Statistically significant differences were determined by one-way ANOVA with Tukey–Kramer multiple comparison method, and different letters (**a** or **b**) indicate significant differences between the six groups (*p* < 0.0001). Source data are provided as a Source Data file. **d** Chimera (ARAC1, RAAC1), axTRPV1, or rTRPV1 was expressed in HEK293T cells and the heat response was analyzed by the whole cell patch-clamp method. Superimposed representative temperature-response relationships are indicated. The threshold for heat-activation of the chimeras (ARAC1, RAAC1), axTRPV1, or rTRPV1 in HEK293T cells were measured as described in the Methods section. The average thresholds were compared (*n* = 6 for axTRPV1 and ARAC1, *n* = 5 for RAAC1, *n* = 4 for rTRPV1). Each data point represents the mean ± SE. Statistically significant differences were determined by one-way ANOVA with Tukey–Kramer multiple comparison method and different letters (**a** or **b**) indicate significant differences between the four groups (*p* < 0.001). Source data are provided as a Source Data file.

(40.81 ± 1.68 °C), demonstrating that the N-terminal end of rTRPV1 can elevate the threshold temperature of axTRPV1. Thus, it is strongly suggested that the N-terminal short region plays a crucial role in determining the threshold temperature of TRPV1.

### Double amino acids substitution in Ankyrin Repeat 1 drastically shifts the TRPV1 thermal threshold

To identify the AA at the N-terminal end of axTRPV1 responsible for LTA, we generated seven additional chimeras between axTRPV1 and rTRPV1 (ARNT1-7) and examined their heat response (Fig. 5). First, we found that only ARNT5, which contains 127–166 AA of axTRPV1,

exhibited LTA (Fig. 5b), whereas both ARNT6 and ARNT7 exhibited LTA. Based on a comparison between the AA sequences of the rat and four tailed amphibian TRPV1 genes, a single mutation at position 128 (Q128R) and a quadruple mutation at position 137–140 (EGDK to QSNC) were introduced into ARNT6. The quadruple mutant exhibited LTA similar to ARNT6, whereas the Q128R mutation canceled the LTA of ARNT6, suggesting an important role for the glutamine residue at position 128 (Fig. 5c). For ARNT7, three single mutations (R151Q, L154K, E160S) were introduced. The R151Q and E160S mutations had little effect on LTA, but the L154K mutant did not exhibit LTA as rTRPV1 (Fig. 5d). These results indicate an important role for the leucine

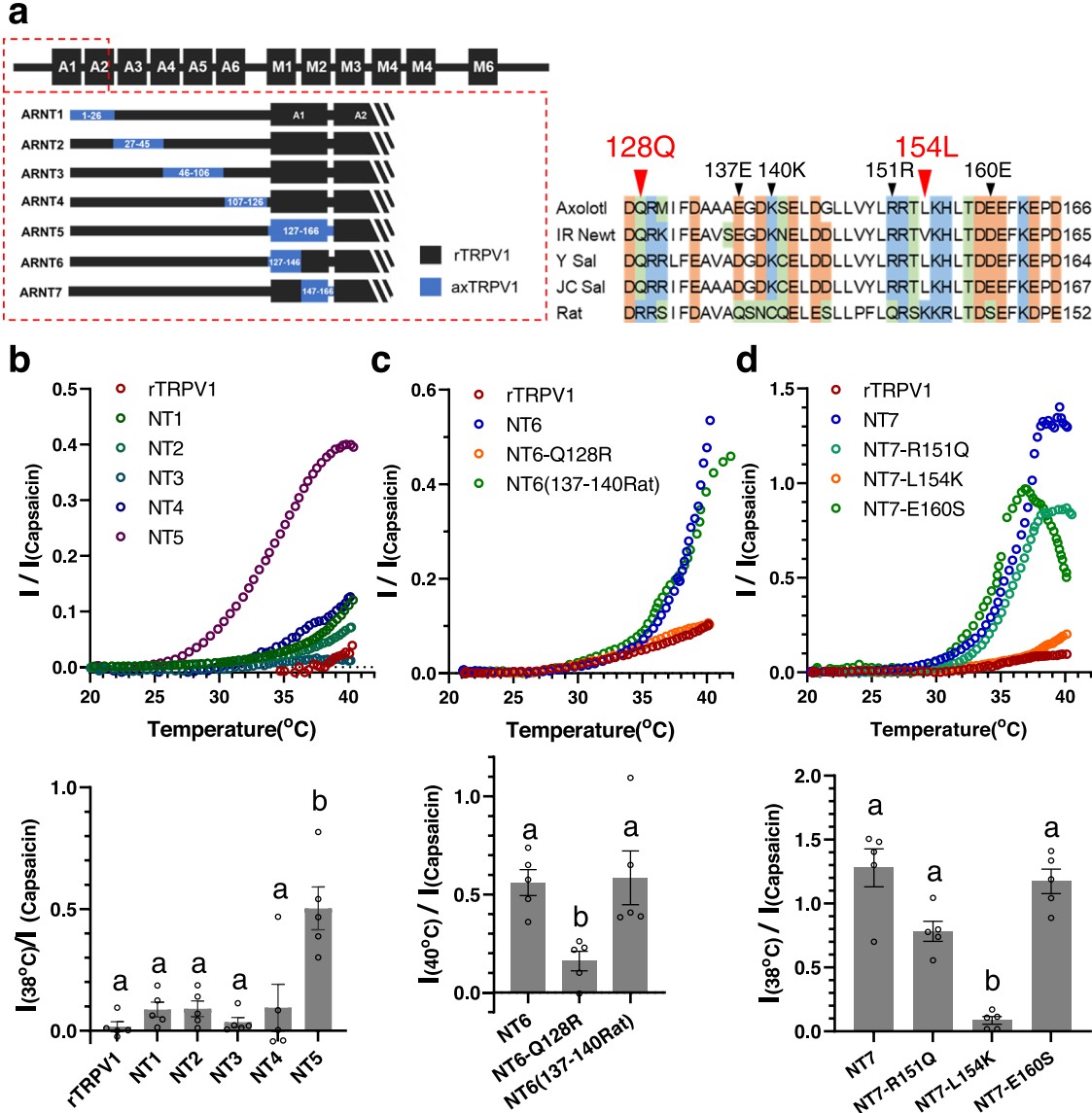

**Fig. 5 | Role of 128Q and 154 L of Ankyrin Repeat 1 of axolotl TRPV1 in reduced thermal threshold. a** Schematic representation of the N-terminal chimeras of axolotl TRPV1. Chimeric channels were generated between rat (black) and axolotl TRPV1 (Blue) on the basis of the location of the ankyrin repeats (A1–A6). Sequence alignment of Ankyrin Repeat 1 of four tailed amphibians and rat TRPV1. **b** Chimera (ARNT1-5) or rTRPV1 was expressed in *Xenopus* oocytes and the representative heat-induced currents were standardized by capsaicin-induced currents at 5 μM are indicated. Average currents for heat stimulation at 38 °C standardized by the capsaicin-induced currents of each chimera in oocytes were compared (*n* = 5). Each data point represents the mean ± SE. Statistically significant differences were determined by one-way ANOVA with Tukey–Kramer multiple comparison method and different letters (**a** or **b**) indicate significant differences between the six groups (*p* < 0.01). Source data are provided as a Source Data file. **c** Q128R mutation was introduced in ARNT6 chimera (NT6-Q128R). The sequence of the 137–140AA (EGDK) of axTRPV1 was exchanged with the corresponding sequence (QSNC) of rTRPV1 in the ARNT6 chimera [NT6(137–140Rat)]. Chimera mutants [ARNT6, NT6-Q128R, NT6(137–140Rat)] or rTRPV1 was expressed in *Xenopus* oocytes and the

representative heat-induced currents standardized by the capsaicin-induced currents at 5 μM are indicated. Average currents for heat stimulation at 40 °C standardized by capsaicin-induced currents for each mutant in oocytes were compared (*n* = 5). Each data point represents the mean ± SE. Statistically significant differences were determined by one-way ANOVA with Tukey–Kramer multiple comparison method and different letters (a or b) indicate significant differences between the three groups (*p* < 0.05). Source data are provided as a Source Data file. **d** A R151Q, L154K, or E160S mutation was introduced into the ARNT7 chimera (NT7-R151Q, NT7-L154K, NT7-E160S). Chimera mutants (NT7, NT7-R151Q, NT7-L154K, NT7-E160S) or rTRPV1 were expressed in *Xenopus* oocytes and the representative heat-induced currents standardized by capsaicin-induced currents at 5 μM are indicated. The average currents for heat stimulation at 38 °C standardized by capsaicin-induced currents for each mutant in oocytes were compared (*n* = 5). Each data point represents the mean ± SE. Statistically significant differences were determined by one-way ANOVA with Tukey–Kramer multiple comparison method and different letters (a or b) indicate significant differences between the four groups (*p* < 0.001). Source data are provided as a Source Data file.

residue at position 154 in the LTA of axTRPV1. Thus, our findings suggest that the residues, Q128 and L154, of ANK1 in axTRPV1 play an important role in its unique heat-activation character and lower temperature threshold.

The corresponding AA residues of rTRPV1 are R114 and K140. To further understand how the amino acid residues at these two positions are involved in determining the threshold for heat-activation, we

created three mutants of rTRPV1 and axTRPV1 and examined their thermal threshold in a *Xenopus* oocyte expression system. For the rTRPV1 mutants, R114 and K140 were mutated to the corresponding Q (128) and L (154) of axTRPV1. Although rTRPV1 was hardly heat-activated up to 40 °C, the two single mutants (R114Q, K140L) exhibited a tendency of heat-activation from a lower temperature. The double mutant (R114Q/K140L) exhibited apparent LTA (Fig. 6a). These results

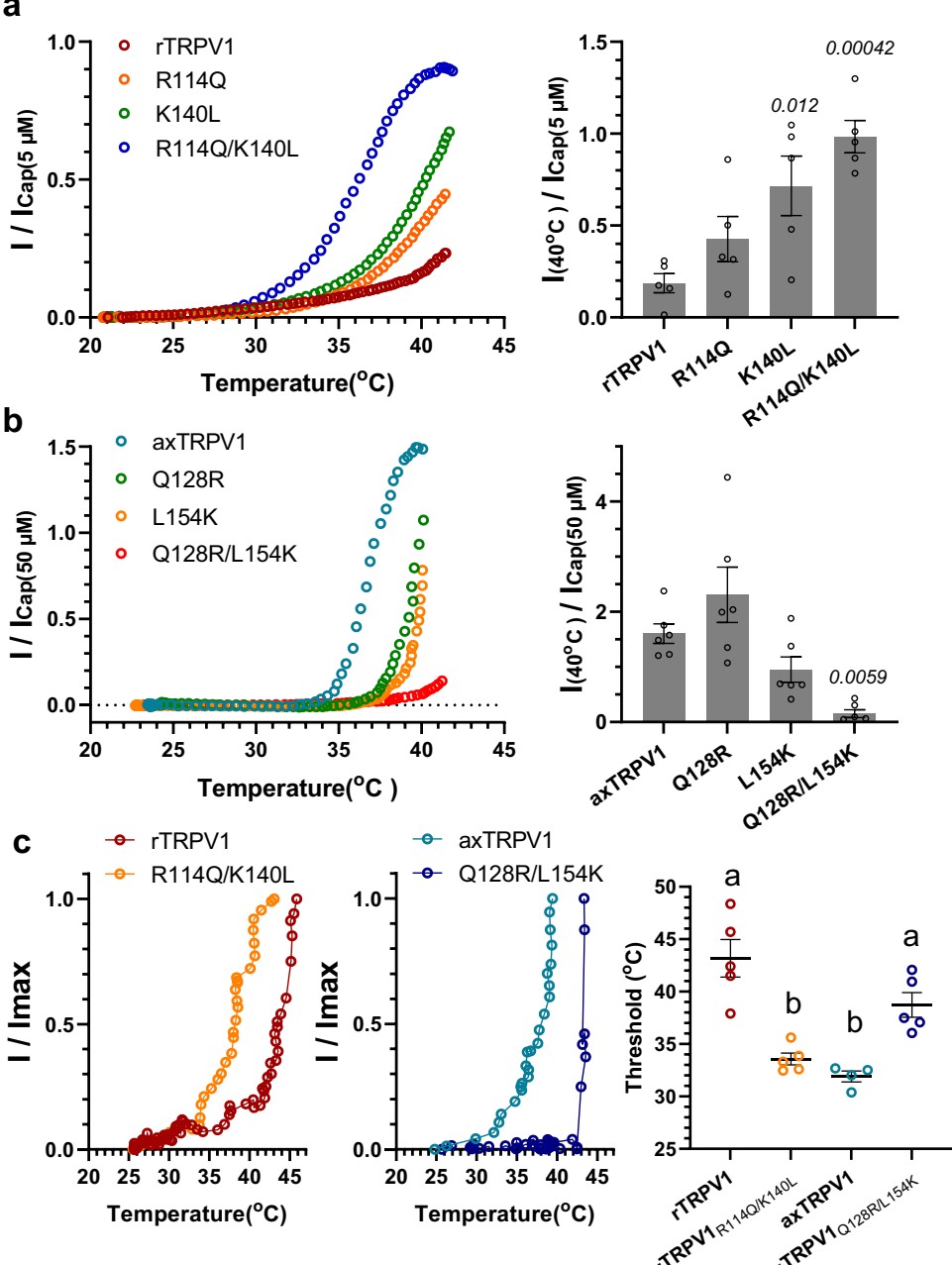

**Fig. 6 | Two amino acids of Ankyrin Repeat 1 determine the heat-activation threshold of axolotl and rat TRPV1s. a** Rat TRPV1 or its mutant (R114Q, K140L, R114Q/K140L) was expressed in *Xenopus* oocytes and the representative heat-induced currents standardized by the capsaicin-induced currents at 5 μM are indicated. Average currents for heat stimulation at 40 °C standardized by capsaicin-induced currents for each chimera in oocytes were compared (*n* = 5). Each data point represents the mean ± SE. Statistical significance for differences with the rTRPV1 group was determined by one-way ANOVA followed by Dunnett's multiple comparison test, and *P* values given in italics indicate statistical significance when <0.05. Source data are provided as a Source Data file. **b** Axolotl TRPV1 or its mutants (Q128R, L154K, Q128R/L154K) was expressed in *Xenopus* oocytes and the representative heat-induced currents standardized by capsaicin-induced currents at 50 μM are indicated. Average currents for heat stimulation at 40 °C standardized by capsaicin-induced currents for each chimera in oocytes were compared (*n* = 6).

Each data point represents the mean ± SE. Statistical significance for the differences with the axTRPV1 group were determined by one-way ANOVA followed by Dunnett's multiple comparison test, and *P* values given in italics indicate statistical significance when <0.05. Source data are provided as a Source Data file. **c** Rat TRPV1 or its double mutant (R114Q/K140L) was expressed in HEK293T cells and the representative heat-induced current response analyzed by the whole cell patch-clamp method was indicated (left). Axolotl TRPV1 or its double mutant (Q128R/ L154K) was expressed and the heat response in HEK293T cells is shown (middle). The average of the thresholds for heat-activation were compared (*n* = 5 for rTRPV1, rTRPV1 double mutant and axTRPV1 double mutant, *n* = 4 for axTRPV1). Each data point represents the mean ± SE. Statistically significant differences were determined by one-way ANOVA with Tukey–Kramer multiple comparison method and different letters (**a** or **b**) indicate significant differences between the four groups (*p* < 0.05). Source data are provided as a Source Data file.

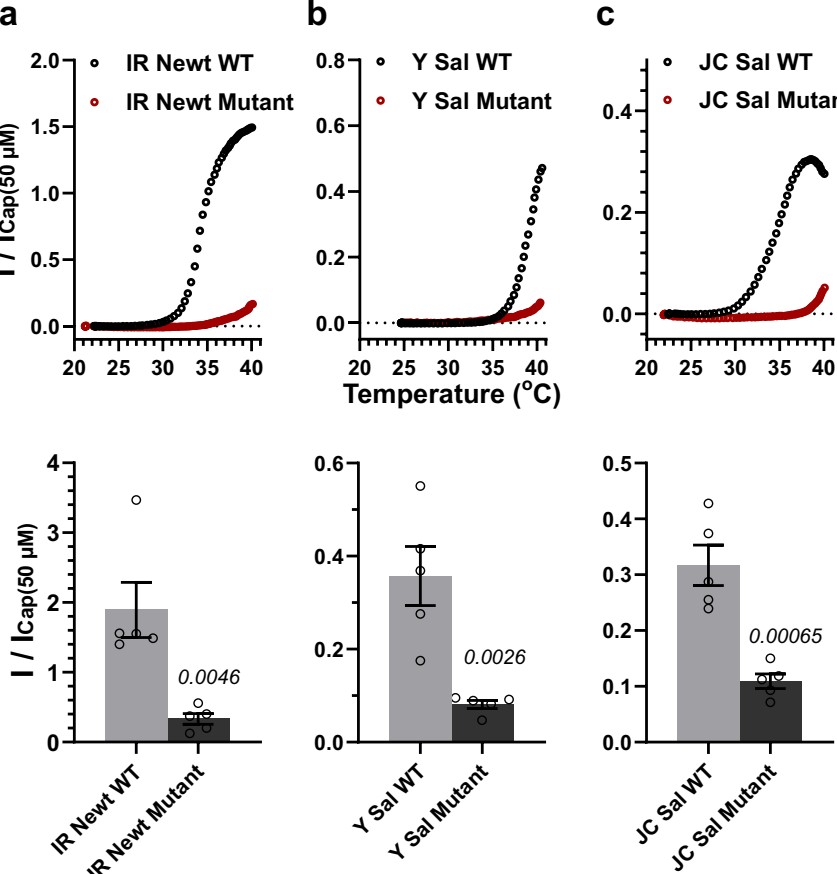

**Fig. 7 | Two amino acids of Ankyrin Repeat 1 determine the reduced thermal threshold of urodelan TRPV1s. a** IR Newt TRPV1 or its mutant (Q127R/V153K) was expressed in *Xenopus* oocytes and the representative heat-induced currents standardized by the capsaicin-induced currents at 50 µM were indicated. Average currents for heat stimulation at 40 °C standardized by capsaicin-induced currents were compared (*n* = 5). Each data point represents the mean ± SE. The statistical significance for the difference was determined by a Student's two-sided unpaired *t*-test and *P* values were indicated in italics. Source data are provided as a Source Data file. **b** Y Sal TRPV1 or its mutant (Q126R/L152K) was expressed in oocytes and representative heat-induced currents are indicated. Average heat-induced currents

at 40 °C were compared (n = 5). Each data point represents the mean ± SE. The statistical significance for the difference was determined by a Student's two-sided unpaired *t*-test and *P* values were indicated in italics. Source data are provided as a Source Data file. **c** JC Sal TRPV1 or its mutant (Q129R/L155K) was expressed in oocytes and the representative heat-induced currents are indicated. Average heat-induced currents at 40 °C were compared (*n* = 5). Each data point represents the mean ± SE. The statistical significance for the difference was determined by a Student's two-sided unpaired *t*-test and *P* values were indicated in italics. Source data are provided as a Source Data file.

clearly indicate that rTRPV1 with a heat-activation threshold of 43 °C can be altered to an LTA type by mutating these two AA.

For the axTRPV1 mutants, Q128Q and L154 were mutated to the corresponding R (114) and K (140) of rTRPV1 (Fig. 6b). The LTA of axTRPV1 was clearly observed; however, the two single mutants (Q128R, L154K) exhibited a less remarkable LTA. The double mutant (Q128R/L154K) was not heat-activated up to 40 °C. Thus, only by mutating these two AA could axTRPV1 be changed to a less sensitive sensor, which is not activated at lower temperatures.

To determine how these double mutations change the threshold temperature for heat-activation of rTRPV1 and axTRPV1, we performed whole cell patch-clamp recording using a HEK293T expression system. As shown in Fig. 6c, rTRPV1 exhibited a threshold temperature for heat-activation of 43.18 ± 1.80 °C, whereas the double mutation of R114Q/K140L markedly reduced it to 33.57 ± 0.57 °C. In contrast, the threshold for axTRPV1 of 31.90 ± 0.52 °C was significantly increased to 38.73 ± 1.17 °C by the reverse double mutation (Q128R/L154K). These results demonstrate that the two AA residues at these two positions, which correspond to the 2nd and 28th position in the ANK1 region of the N-terminal domain (R114 and K140 of rTRPV1, Q128 and L154 of axTRPV1), are involved in determining the threshold for heat-activation.

TRPV1s from three tailed amphibians other than axolotls (IR Newt, Y Sal, JC Sal) also showed a lower threshold temperature for heat-activation (Fig. 3). The AAs at the positions corresponding to 128Q and 154 L of axTRPV1 are well conserved among the tailed amphibians (Fig. 5a). To identify the shared mechanism which contributes to the heat-sensing characteristics of TRPV1s of IR Newt, Y Sal, and JC Sal, the same double mutation (to R and to K) at the corresponding positions to those of axTRPV1 was introduced into each TRPV1 of the tailed amphibians and the heat-activation properties were analyzed (Fig. 7). The results indicated that this double mutation also eliminated LTA for all three tailed amphibian TRPV1s.

## Molecular mechanism for the drastic shift of TRPV1 thermal threshold by amino acid substitution in Ankyrin Repeat 1

The double mutations from uncharged to basic AAs (Q128R/L154K) of axTRPV1 elevated the threshold temperature for heat-activation. Therefore, we hypothesized that the generation of an electrostatic interaction between the three-dimensional adjacent acidic AAs may be involved in the shift of the heat threshold to a high temperature. To search for candidate acidic AAs, a 3D protein structural model of axTRPV1 was generated by homology modeling using the cryo-electron microscopy (EM) structures of full-length mouse TRPV3 and

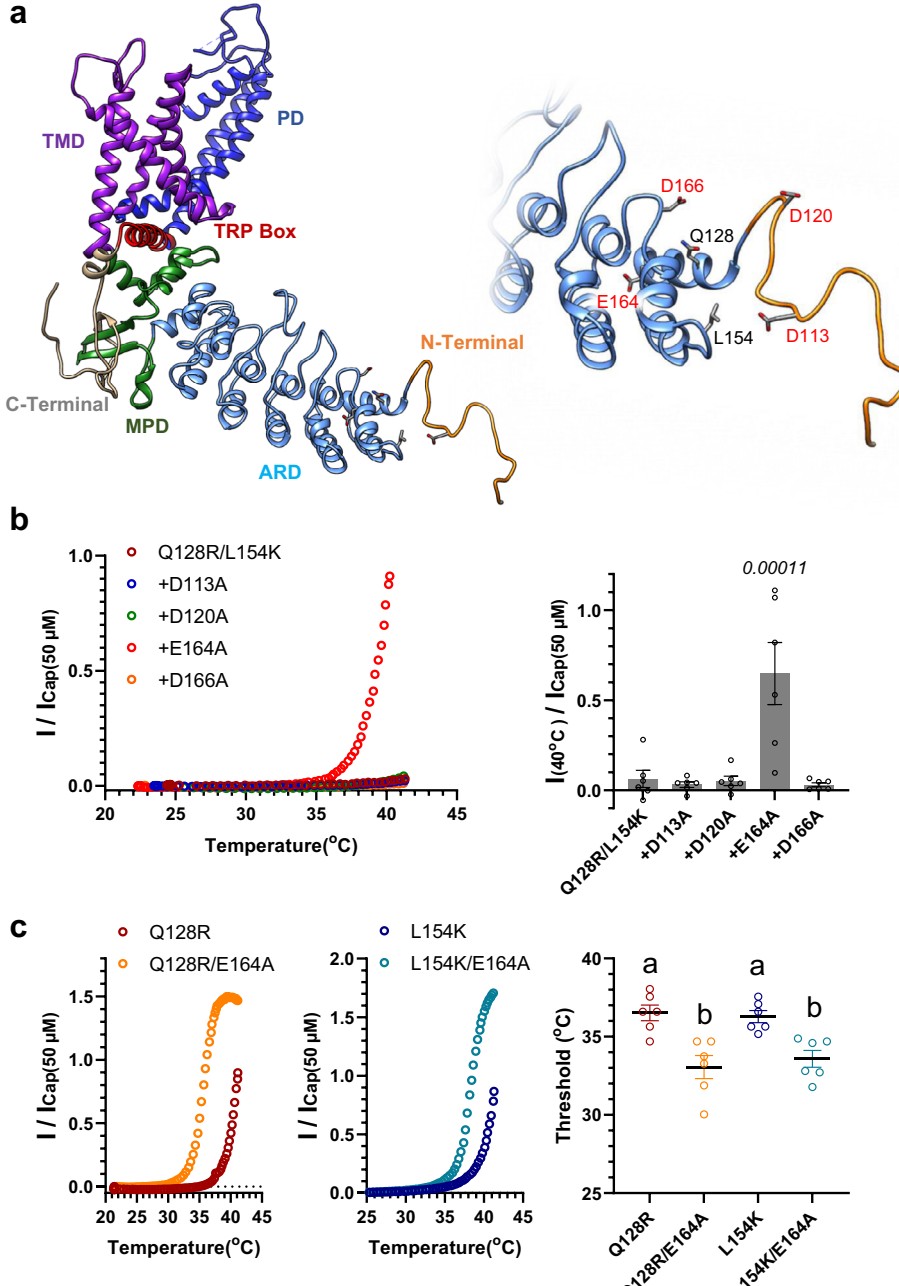

**Fig. 8 | Search for acidic amino acids which electrostatically interact with mutated basic amino acids (Q128R/L154K) of axTRPV1. a** The 3D protein structural model of axTRPV1 was generated by homology modeling with the cryo-electron microscopy structures of full-length mouse TRPV3 (7MIK) and rTRPV1 (7LP9). **b** In addition to the Q128K/L154K mutation, another mutation (D113A, D120A, E164A, or D166A) was introduced in axTRPV1. Each mutant was expressed in *Xenopus* oocytes, and the representative heat-induced currents standardized by capsaicin-induced currents at 50 μM are indicated. Average currents for heat stimulation at 40 °C standardized by capsaicin-induced currents were compared ($n = 6$). Each data point represents the mean ± SE. Statistically significant differences for the Q128R/L154K group were determined by one-way ANOVA followed by Dunnett's multiple comparison test, and *P* values given in italics indicate statistical significance when <0.05. Source data are provided as a Source Data file. **c** In addition to the single mutation of Q128K or L154K, the E164A mutation was introduced in axTRPV1. Each mutant was expressed in oocytes and the representative heat-induced currents are indicated. The average threshold for heat-activation for each mutant was compared ($n = 6$). Each data point represents the mean ± SE. Statistically significant differences were determined by one-way ANOVA with Tukey–Kramer multiple comparison method and different letters (**a** or **b**) indicate significant differences between the four groups ($p < 0.05$). Source data are provided as a Source Data file.

rTRPV1. The mouse TRPV3 structure was also used as a template because the number of unmodeled atoms was smaller than that of TRPV1. As shown in Fig. 8a, we focused on four acidic AAs, D113, D120, E164, and D166, that may be close to Q128 and L154 in space. To examine the presence of the electrostatic interaction with a mutated basic AA, each of these acidic AAs was changed to alanine (D113A,

D120A, E164A, D166A) in the double mutant of axTRPV1 and heat-activation of the four mutants was determined (Fig. 8b). Heat stimulation to approximately 40 °C activated one mutant of E164A, indicating that the E164A mutation partly cancels the mutation effect of Q128R/L154K on axTRPV1. To further determine whether E164 interacts with R128 or K154, we determined the effects of E164A mutation

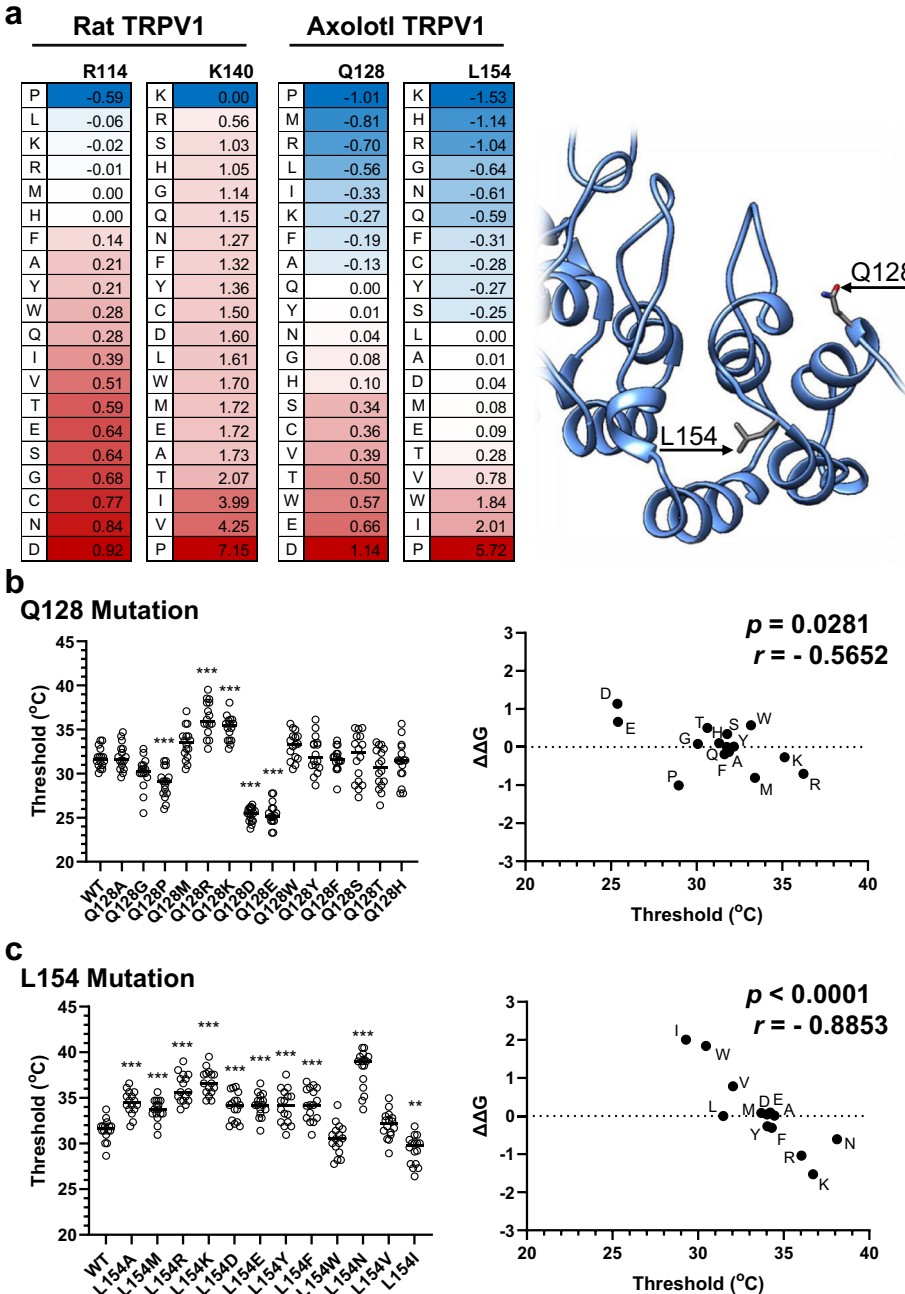

**Fig. 9 | The mutational folding energy changes and heat-activation thresholds of Ankyrin Repeat 1 mutants of axTRPV1. a** The protein 3D structure model of rTRPV1 or axTRPV1 was generated by homology modeling with the cryo-electron microscopy structure of full-length rTRPV1 (7LP9). The mutational folding energy changes (*ΔΔG*) were evaluated with FoldX. The structure of the N-terminus of axTRPV1 is shown. **b** In the position of Q128 of axTRPV1, a mutation to change the amino acid residue was introduced. Each mutant was expressed in *Xenopus* oocytes, the heat-induced currents were analyzed, and the threshold temperature for heat-activation was determined. The average threshold for each mutant was compared (*n* = 16). The mean ± SE was indicated. Statistically significant differences were determined for the axTRPV1 (WT) group by one-way ANOVA followed by Dunnett's multiple comparison test and indicated by *** (*p* < 0.001). Correlation between *ΔΔG* and the threshold temperatures of the mutants was assessed using Pearson's correlation. Source data are provided as a Source Data file. **c** For the position of L154 of axTRPV1, a mutation to change the amino acid residue was introduced. Each mutant was expressed in oocytes and the average threshold temperature for heat-activation was compared (*n* = 16). The mean ± SE was indicated. Statistically significant differences were determined for the axTRPV1 (WT) group by one-way ANOVA followed by Dunnett's multiple comparison test and indicated by ** (*p* < 0.01) and *** (*p* < 0.001). The correlation between *ΔΔG* and the threshold temperatures of the mutants was assessed using Pearson's correlation. Source data are provided as a Source Data file.

on the single mutants, Q128R and L154K. In both cases, this mutation shifted the thermal threshold for heat-activation to a lower temperature (Fig. 8c). The results indicate that the E164A mutation may allow axTRPV1 to open from a lower temperature independently through the Q128R and/or L154K mutations.

Next, 3D protein structural models for rTRPV1 and axTRPV1 were further generated by homology modeling using the cryo-EM structure of the full-length rat TRPV1 and changes in the Gibbs free energy of the protein (*ΔΔG*) were predicted. In other words, the protein stability caused by a mutation using FoldX (Fig. 9a). In the rTRPV1 model, the

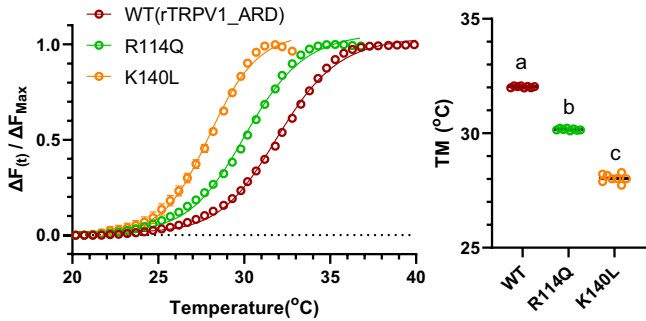

**Fig. 10 | Mutation in ANK1 shifts the heat-dependent conformal change of the N-terminus of rTRPV1.** The proteins of the N-terminus (100-362aa) from wild rTRPV1 and two mutants (R114Q, L140L) were purified and used for thermal shift unfolding assay. Data was obtained every 0.5 °C. $\Delta F_{(t)}/\Delta F_{max} = F_{(t)} - F_{(20\,°C)}/F_{max} - F_{(20\,°C)}$. The melting temperature, $T_m$ was obtained. The mean ± SE was indicated. The average $T_m$ values were 32.03 ± 0.01 °C for WT, 30.16 ± 0.02 °C for R114Q, and 28.04 ± 0.06 °C for K140L ($n = 8$). Statistical significance of the differences was tested by one-way ANOVA with Tukey–Kramer multiple comparison method and different letters (**a–c**) indicate significant differences between the three groups ($p < 0.0001$). Source data are provided as a Source Data file.

R114Q and K140L mutations yielded positive values, indicating that the N-terminus of rTRPV1 protein may become less stable by these mutations. In the axTRPV1 model, values for the mutations of Q128R and L154K were negative, indicating that the N-terminus of axTRPV1 protein may become more stable by these mutations. Recently, Kwon et al. proposed a heat-dependent opening model for rTRPV1 based on cryo-EM structures[16]. They demonstrated that the heat-induced conformational transition may occur in a stepwise manner. The first transition involves increased flexibility in the N-terminal ANKs and then the contraction of all subdomains (ANKs, the coupling domain, the TRP helix, the S1–S4 domain). Then, in the second transition, local conformational changes of the outer pore and S6 result in dilation of the selective filter and the S6 gate to open the channel. In our experiments, the mutation that may decrease the stability of the N-terminus of TRPV1 protein may lower the threshold temperature of heat-activation, and conversely, the mutation which may increase the protein stability of the N-terminus may elevate the threshold. Taken together with the heat opening model of TRPV1 proposed by Kwon et al., it is possible that triggering the first transition may be the basis for determining the threshold temperature.

Based on this hypothesis, we determined the activation-threshold temperature for 13 additional mutants at Q128 and 11 mutants at L154 of axTRPV1 (Fig. 9b, c). For Q128 of axTRPV1, the threshold temperature was significantly inversely-correlated with the value of $\Delta\Delta G$ ($r = -0.5652$, $p = 0.0281$). For the position at L154, we further observed a stronger inverse correlation with $\Delta\Delta G$ ($r = -0.8853$, $p = 0.0001$). These analyses clearly indicate the apparent tendency of the mutant with a higher $\Delta\Delta G$ exhibiting a less stable structure and a lower threshold temperature, whereas the mutant with a lower $\Delta\Delta G$ has an elevated threshold. The results indicate that the protein structure stability of the N-terminus, which is efficiently controlled by a single AA at Q128 and L154 on axTRPV1, may determine the heat-activation threshold of axTRPV1. This suggests a mechanism for the drastic shift of the TRPV1 thermal threshold by amino acid substitution in ANK1.

### The mutation identified by the electrophysiological experiments also changes the thermal stability of the purified N-terminus protein containing ANKs of TRPV1

Recently, using biochemically purified N-terminus protein containing whole ANKs from TRPV1 (100–362aa) in solution, the thermal shift unfolding with SYPRO orange, circular dichroism, and tryptophan fluorescence measurements were performed and it was revealed that

the N-terminus undergoes apparent structure changes in accordance with an increase in the temperature that may lead to the TRPV1 channel activation[14]. Our electrophysiological data demonstrated that R114Q and K140L mutations in ANK1 in the N-terminus of rTRPV1 significantly lowered changed the heat-activation threshold (Fig. 6a, b). Therefore, we examined whether R114Q and K140L mutations really affect the thermal stability of the purified rTRPV1 N-terminus protein by the thermal shift unfolding (Fig. 10). The melting temperature, Tm for the wild-type N-terminus was 32.03 ± 0.01 °C, which is close to the reported value[14]. Two mutations significantly decreased Tm values of the thermal shift unfolding of the N-terminus (R114Q: 30.16 ± 0.02 °C, K140L: 28.04 ± 0.06 °C), indicating that the thermal stability was downregulated by the mutation (Fig. 10). These results demonstrate that the protein structure stability of the N-terminus is efficiently controlled by two amino acid residues at the 2nd and 28th position of ANK1 in TRPV1, and strongly suggest that the ANK1 functions as a structural module which determines the heat-activation threshold of TRPV1.

## Discussion

Previously, we characterized TRPV1 from axolotls[18]. In the present study, we cloned TRPV1s from three tailed amphibians, IR Newt, Y Sal, and JC Sal, and determined their heat activating property. The threshold temperature for heat-activation of urodelan TRPV1s were all lower compared with the threshold of 43 °C for rTRPV1. To understand the structure of axTRPV1 for LTA, we generated various chimera constructs of axTRPV1 and rTRPV1. We found that ANK1 was composed of 40 AAs in the N-terminus of axTRPV1 and is necessary for LTA. When Q128 and L154 in ANK1 of axTRPV1 were changed to the corresponding AA residues of rTRPV1 of R and K, LTA was abolished and the threshold was elevated. Furthermore, a reverse double mutation of rTRPV1 (R114Q/K140L) reduced its threshold temperature by approximately ten degrees. Because the AAs at the position corresponding to 128Q and 154 L of axTRPV1 are conserved among tailed amphibians, the same double mutation (to R and to K) on TRPV1s of IR Newt, Y Sal, and JC Sal at the same positions as axTRPV1 was introduced. The results indicated that this double mutation canceled LTA for all urodelan TRPV1s. This indicates that the properties of the AAs at these two positions in ANK1 are necessary for the heat activating property of TRPV1. Furthermore, the presence of the Q and L/V residues at these positions determines the lower threshold temperature for heat-activation of tailed amphibian TRPV1s.

Next, we generated a 3D protein model of axTRPV1 using homology modeling and predicted the changes in Gibbs free energy of a protein caused by a mutation at 128Q or 154 L for comparison with the experimentally determined thresholds of heat-activation for the mutants. At both sites, we detected a significant inverse correlation between the energy change and the threshold temperature, and proposed a possible mechanism regarding the regulation of the thermal threshold of axTRPV1. It is that the protein stability of the N-terminus, which is efficiently controlled by the property of a single AA at Q128 and L128 on axTRPV1, may determine the heat-activation threshold of axTRPV1.

Previously, using the biochemically isolated ANK domain of rTRPV1, the temperature-dependent dynamics of protein conformation were observed[14], and a possibility that the ANK domain of TRPV1 may function as a structural module which contributes to the control of the temperature sensitivity was demonstrated. Therefore, we prepared the purified protein of the N-terminus containing the ANKs of TRPV1 wild type and mutants, and analyzed the conformation, thermal stability, and effects of mutations on protein stability and heat-dependency. Results clearly demonstrated that the protein structure stability of the N-terminus is efficiently controlled by two residues at the 2nd and 28th position of ANK1 of TRPV1, and strongly suggests that the ANK1 functions as a structural module determining the heat-activation threshold of TRPV1.

Earlier studies proposed that the pore region of the channel molecule is engaged in the heat-induced activation of TRPV1[11,24–27]. Furthermore, AA substitutions at three positions in the pore may participate in tuning the heat-activated threshold by 2 °C in fruit bat TRPV1[12]. Both N- and C- termini, however, have also been reported to be implicated in thermal activation[6,7,9,13]. In particular, it is known that C-terminal truncation of TRPV1 shifts the thermal threshold to a lower temperature[6,8]. In previous studies on the role of the N-terminus, it was demonstrated that the membrane-proximal domain located between the ANKs and the transmembrane domains determines temperature dependence (activation enthalpy) by analyzing TRPV1-TRPV2 chimeras[9]. Thus, it has been suggested that determinants of the heat-sensing module may not be localized at a single channel element, and that topologically distant domains could be coupled to induce the heat-dependent opening of channels. Based on cryo-EM structures, Kwon et al. proposed the heat-dependent opening model of rTRPV1[16]. In this model composed of "Close, Intermediate, and Open", the first transition involves the increased flexibility of the N-terminal ANKs followed by contraction throughout the molecule. The second transition induces local conformational changes of the outer pore and S6 to open the channel. Since the heat-dependent first transition must initiate over the threshold temperature, the process of the increase in flexibility of the N-terminal ANKs turns to determine the heat-activation threshold of TRPV1. Our results indicated that two AA substitutions at the 2nd and 28th position of ANK1 at the N-terminus markedly and reversibly changed the threshold temperature for heat-activation of TRPV1. We further showed that the protein structure stability of the N-terminus is efficiently controlled by two residues at the same two positions of ANK1 of TRPV1, thus, supporting our proposed mechanism that two AA at the 2nd and 28th position of ANK1 determines the thermal threshold of TRPV1 by controlling the heat-dependent increase in flexibility of N-terminus.

Other mutations at different AA positions in N-terminal ANKs, which can affect temperature-dependent protein stability, may also change the heat-activation threshold of TRPV1. Indeed, in a study of heat tolerance in squirrels and camels, it has been reported that two AA changes found in ANK1-3 of squirrel and camel TRPV1s caused resistance to heat-activation at approximately 44 °C[13]. These AA changes (S124N, Q188E) may increase the protein stability of the N-terminus of TRPV1 and the activation of TRPV1 may become less sensitive to heat. We calculated ΔΔG for the 124S and 188Q mutants in the same rTRPV1 model (Supplementary Fig. 4). Both mutants (S124N and Q188E) exhibited negative ΔΔG values, indicating a more stable structure. Because their heat-activation thresholds are considered elevated, the heat tolerance of squirrels and camels may be explained by our hypothesis. A similar mechanism is known for a bacterial voltage-gated sodium channel, in which temperature-induced unfolding of a part of the C-terminus drives channel opening[28].

The heat-activation threshold of urodelan TRPV1s were all apparently lower compared with the sensitivity threshold of rTRPV1. When each threshold temperature of urodelan TRPV1s was compared with the animal behavior to heat, some discrepancies were observed. Noxious responses to heat for ax and Y Sal were observed at 33 °C and 36 °C, respectively, and these behaviors may be explained by their TRPV1 thresholds (30.91 °C ± 0.34 °C, 34.11 °C ± 0.35 °C). For IR Newt, however, the temperature at 36 °C, which is higher than its threshold for TRPV1 (31.57 °C ± 0.36 °C), did not induce a noxious response in the animals. The heat-induced behavior was observed at 39 °C. In the sensory neurons of the IR Newt, the function of TRPV1 may be modified by some unknown mechanism or another thermosensor molecule, thus the threshold temperature of 39 °C may be dominantly expressed. Although JC Sal TRPV1 was heat-activated at a threshold temperature of 27.87 °C ± 0.31 °C, noxious heat behavior was observed from 20 °C. Therefore, sensory neurons of JC Sal must express a heat sensor that is activated at 20 °C. One possible candidate is TRPA1. The

heat-activation characteristics of TRPV1 and TRPA1 from *Xenopus laevis* and *Xenopus tropicalis* have been reported. *X. laevis* is much more sensitive to heat stimulation at a behavior level and the difference in the heat-activation threshold was associated with TRPA1 function[29].

In summary, we indicated that tailed amphibians express TRPV1s with a low temperature threshold for activation, which may have occurred by the substitution of two AA in the ANK1 region to promote cool-habitat selection. Such a simple plasticity of TRPV1 may provide a great advantage to the evolution and adaptation of various animals.

## Methods

### Experimental animals
All of the animal experiments described below were approved by the Animal Experiment Committee of Nagahama Institute of Bio-Science and Technology and were performed based on their guidelines. *Xenopus* oocyte preparation has been described previously[30].

### Behavioral experiments
Each tailed amphibian was placed into a metallic tray (15 cm × 21 cm) with 500 ml of chlorine-free tap water pre-warmed at various temperatures. We monitored the locomotor activity of the animals using a video camera. After placing into pre-warmed tap water, heat-induced movements per 2 s or 1 s were measured over 20–60 s and the mean movement (cm) per 1 min was calculated.

### Cloning, mutagenesis, and plasmid of TRPV1 cDNAs
Molecular cloning of axTRPV1 was done as previously described[18]. We further isolated TRPV1 cDNAs from Iberian ribbed newt (IR Newt, *P. waltl*), Yamato salamander (Y Sal, *H. vandenburghi*), and Jananese clawed salamander (JC Sal, *O. japonicus*). Total RNA was isolated from the brain of the IR Newt, Y Sal, and JC Sal. Reverse-transcribed cDNA (RT) was prepared with random primers and the product was used as a template for PCR amplification. To isolate partial DNA fragments of IR Newt TRPV1 cDNA, we synthesized the following primers based on the EST information (M4959883, M2066090).

F Primer: GGGCAGATGGAGAGTTCTTC
R Primer: TCCCTCCAATCACGGTTATG

Once the expected partial sequence was obtained, the sequence information of the full-length cDNA of IR Newt TRPV1 was determined by the RACE (Rapid Amplification of cDNA Ends) method using GeneRacer™ kit (ThermoFisher Scientific, Waltham, MA) (accession number: LC728477). To isolate Y Sal TRPV1 cDNA, we designed the following two sets of primers based on the conserved region between axTRPV1 and IR Newt TRPV1 cDNAs.

1st F Primer: GCTGGAAGATATTGCCAACAAGAAAG
1st R Primer: ATTCGGGGTCTCGCTGCTGTAG
Nested F Primer: CTTGCTGCAAAGACAGGCAAGAT
Nested R Primer: TGATCTCCAGCACCGAATTGTTC

Using these primers, we amplified a partial DNA fragment of Y Sal TRPV1 cDNA. After confirmation of the sequence of the amplified DNA, the full-length sequence of the cDNA for Y Sal TRPV1 was determined by RACE (accession number: LC728478). To isolate JC Sal TRPV1 cDNA, we designed the following two sets of primers corresponding to the two conserved regions between axTRPV1 and IR Newt TRPV1 cDNAs.

1F Primer: TTGCTGCAAAGACAGGCAAG
1R Primer: AAGATGCGCTTGACAAATCG
2F Primer: CAGGAAAGCAAGAGCATCTG
2R Primer: TCCACTCTGAAGCACCATCTG

After confirmation of the amplified DNA sequence, we synthesized the following third set of primers corresponding to the internal sequence of each amplified DNA.

3F Primer: TCTGTCCCGCAAGTTCACAG
3R Primer: TCAGCGAGTTCAGGAAGCTC

Using these primers, we successfully amplified a partial DNA fragment of JC Sal TRPV1 cDNA. Then, the full-length sequence of

cDNA of JC Sal TRPV1 was determined using the RACE method (accession number: LC728479). For *Xenopus* oocyte expression, each urodelan TRPV1 cDNA was cloned into pGEMHE, which included the 5′ and 3′ non-coding sequence of the *Xenopus* β-globin gene[31]. Chimeras and point mutations were generated by overlapping PCR and the KOD -plus- Mutagenesis Kit (TOYOBO, Osaka, Japan). All constructs were verified by sequencing.

### Oocyte electrophysiology

An outline of the methods used for the electrophysiological experiments using *Xenopus* oocytes was described previously[32]. After injecting 50 nl of TRPV1 cRNA (0.1 µg/µl), oocytes were incubated in frog Ringer solution at 17 °C for 3–4 days. Ionic currents were recorded by the two-electrode voltage clamp method. The recording bath solution contained 96 mM NaCl, 2 mM KCl, 3 mM MgCl$_2$, 5 mM HEPES, and 2 mM NaOH at pH 7.4. Oocytes were voltage-clamped at −20 mV. The current data was obtained using 400 ms step pulse from −80 mV for 100 ms to +40 mV for 100 ms and applied every 1 s. For heat stimulation, the bath solution was heated with an in-line heater controller (CL-100, Warner Instruments, Holliston, MA) and was applied by perfusion. Induced currents at −80 mV were analyzed. Temperature thresholds and $Q_{10}$ were determined as described previously by Gracheva et al.[33]. Temperature thresholds represent the point of intersection between linear fits to baseline and the steepest component of the Arrhenius profile. The Arrhenius curve was obtained by plotting the current at −80 mV on a log-scale against the reciprocal of the absolute temperature. To obtain the current-voltage relationship, voltage ramps from −100 mV to +100 mV were applied every 2 s.

### HEK293T electrophysiology

Macroscopic currents were recorded in HEK293T cells (human embryonic kidney cell line) transfected with the TRPV1 constructs along with a fluorescent protein marker as previously described[34]. HEK293T cell, established by Professor David Baltimore, is used in this study. The HEK293T cell was gifted from Dr. Masaki Iizuka (in Nippon Boehringer Ingelheim Co. Ltd. Pharma Research Institute) to Dr. Yoshihiro Kubo. Whole cell patch-clamp recordings were performed 24–72 h after transfection with Axopatch 200B amplifiers and pClamp 9 software (Molecular Devices, San Jose, CA). Glass pipettes were filled with an internal solution consisting of 130 mM KCl, 5 mM Na$_2$-ATP, 0.5 mM EGTA, 0.1 mM CaCl$_2$, 10 mM HEPES, 4 mM MgCl$_2$, 0.3 mM GTP, pH 7.4, adjusted with KOH. The bath solution consisted of 140 mM NaCl, 4 mM KCl, 10 mM HEPES, 0.3 mM MgCl$_2$, pH 7.4. The whole cell configuration was established in the bath solution, and the cell was then held at −20 mV and perfused with nominally Ca$^{2+}$ free (0 mM CaCl$_2$) bath solution (or the ice cold Ca$^{2+}$ free bath solution when the low temperature threshold constructs were tested). The macroscopic current was then recorded using the step pulse protocol above and the temperature of the bath solution was simultaneously monitored using a temperature controller (TC324B, Warner Instruments). Heat stimulation was performed by perfusion of the heated Ca$^{2+}$-free bath solution passing through the in-line solution heater (SH-27B, Warner Instruments) and connected to the TC324B.

### Molecular modeling

Initial structural models of axTRPV1 and rTRPV1 were constructed using MODELER 10.1[35], based on the cryo-EM structures of full-length mouse TRPV3 (PDB ID:7MIK)[15] and rat TRPV1 (PDB ID: 7LP9)[16]. axTRPV1 and rTRPV1 were remodeled from the identical experimental structure, 7LP9, because a considerable number of the side chains were unmodeled. The mouse TRPV3 structure, 7MIK, was also employed as a template because the number of unmodeled atoms was smaller than that of 7LP9. The models were further refined by iteratively applying molecular dynamics and the geometry minimization procedures of PHENIX 1.19.2[36]. The model quality was evaluated using MOLPROBITY

4.5[37] and the percentages of rotamer outlier, Ramachandran outlier, and crash score of the final models based on 7LP9 were, 3.7, 0.0, and 3.8 for axTRPV1, and 2.8, 0.0, and 3.8 for rTRPV1, respectively. Those based on 7MIK were 35.9, 0.2, and 10.0 for axTRPV1, and 29.6, 0.0, and 11.9 for rTRPV1, respectively. The mutational folding energy changes ($\Delta\Delta G_{mut}$) were evaluated with FOLDX 4[38].

### Expression of N-terminus containing ANKs and purification

The DNA sequence encoding for the N-terminus containing all ANKs (residues 100–362) of rat rTRPV1, R114Q, or K140L was cloned into a pQE30 vector (QIAGEN, Hilden, Germany). For expression, the *Escherichia coli* JM109 (NIPPON GENE, Tokyo, Japan) was transformed with this vector. Batch cultures of transformed cells were grown in Luria Bertani broth (Nacalai Tesque, Tokyo, Japan) to an OD of 0.7–0.8, and protein expression was then induced with 0.1 mM isopropyl-β-D-1-thiogalactopyranoside (Nacalai Tesque, Tokyo, Japan) for 40–48 h at 15 °C. The cell pellet was lysed in 50 mM Tris (pH 8.0), 0.1 mM dithiothreitol (Invitrogen, Carlsbad, CA), and 300 mM NaCl (Wako, Miyazaki, Japan). The lysate was purified by Ni-NTA Agarose (QIAGEN, Hilden, Germany) with an imidazole gradient (10, 20, 50, 100, 200, 300 mM). The eluted protein was desalted in a NAP25 column (GE Healthcare, Buckinghamshire, UK) against a 50 mM Tris (pH 8) buffer.

### Thermal shift unfolding assay

Assays were prepared in a 96-well plate for real-time PCR; each well was prepared with 2 µL of SYPRO Orange 50X (Invitrogen, Carlsbad, CA), 18 µL of 0.1 mg/mL each solution of biochemically purified protein of rTRPV1 N-terminus (WT, R114Q, K140L). A real-time thermocycler Mini Opticon (Bio Rad, Hercules, CA) was used for collecting data from 10 to 95 °C, sampling every 0.5 °C. The data were exported and analyzed in Opticon Monitor (Bio Rad, Hercules, CA) and Graph Pad Prism9.2.0 (GraphPad Software, La Jolla, California).

### Statistical analyses

The values and error bars shown in the figures indicate the mean and standard errors. The statistical significance for the difference of two groups was determined by a Student's two-sided unpaired *t*-test and *P* values were indicated in italics. For multiple comparisons, we performed Dunnett's test or the Tukey−Kramer method. When simply compared with the control group, statistical significance for the difference with the control group was determined by one-way ANOVA followed by Dunnett's multiple comparison test, and *P* values given in italics indicate statistical significance when <0.05. For other multiple comparison, statistical significance for the difference was determined by one-way ANOVA with Tukey−Kramer multiple comparison method. Significant differences were indicated by different alphabets such as a, b, and c. Correlations were assessed by Pearson's correlation coefficient.

### Reporting summary

Further information on research design is available in the Nature Portfolio Reporting Summary linked to this article.

## Data availability

The data that support this study are available from the corresponding authors upon request. The source data underlying figures are provided as a Source Data file. Sequencing data that support the findings of this study have been deposited in the DNA Data Bank under the accession number LC728477 (IR Newt TRPV1), LC728478 (Y Sal TRPV1), and LC728479 (JC Sal TRPV1). Source data are provided with this paper.

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

## Acknowledgements

We thank Dr. D. Julius (UCSF, CA, USA) for the cDNA of rat TRPV1. We also thank the National Bio-Resource Project of the Ministry of Education, Culture, Sports, Science and Technology in Japan (MEXT) for providing Iberian ribbed newts. We also    thank Mr. Yuki Shintani for the support of behavior experiments of JC Sal. Photos of urodelans were kindly pro-vided by Mr. Shintaro Seki, a nature photographer. This work was sup-ported by JSPS KAKENHI (Grant Nos. 16K07305 and 22K06324 to O.S.), the Sasakawa Scientific Research Grant from the Japan Science Society (to S.H.), and a grant from the Cooperative Study Program of National Institute for Physiological Sciences (to O.S.).

## Author contributions

S.H. and O.S. conducted the majority of experiments including cloning, mutagenesis, electrophysiological, biochemical, and animal assays. M.T. and Y.K. contributed to the patch-clamp recordings. T.S., con-tributed to the molecular modeling. O.S., S.H., T.S. and Y.K. prepared the manuscript. O.S. conceived and supervised the project.

## Competing interests

The authors declare no competing interests.
