## [Peer Review File · Nature Communications]

Two Single-Point mutations in Ankyrin Repeat One Drastically Change the Threshold Temperature of TRPV1Reviewers' Comments:

Reviewer #2:

Remarks to the Author:

Summary: The manuscript by Hori et al. used a comparative approach to discover new aspects of the thermal sensitivity of TRPV1 channels at the molecular level. The author compared the thermal sensitivity of TRPV1 cloned from rats with several amphibian species that prefer cooler environments. These studies revealed that TRPV1 channels cloned from amphibians were much less sensitive to capsaicin activation than rats. Amphibian channels were more sensitive to activation by acidic pH, and the activation temperature for amphibian TRPV1 channels was approximately 10 C lower than that of rat TRPV1 channels. Chimeric and sequence analysis revealed two amino acid residues within the first N-terminal ankyrin repeat domain (ANK1), Q128 and L154, conserved among amphibians but differed from rats. Substitution of these residues into the rat sequence lowered the temperature required for activation and increased sensitivity to capsaicin, and single substitutions had only a small effect. A 3D-modeling approach was used to predict critical residues within ANK1 that contribute to thermal sensitivity. The authors propose that amphibian ANK1 structures containing Q128 and L154 are less stable compared with the same domain in rats, leading to lower temperature activation.

Comments for the authors:

This is a fascinating and well-executed study. The data are presented clearly, and my only suggestion for improvement is to reorganize and improve the discussion. In my opinion, this section is somewhat rambling and distracts from an otherwise solid manuscript.

Reviewer #3:

Remarks to the Author:

This manuscript is a follow up on previous works describing a shift in temperature threshold of activation in axolotl TRPV1 channels (Hori, S., & Saitoh, O. BBRC; 2020). In the present work Hori and cols describe an interesting functional-structural observation, that two specific residues located in the first ankyrin repeat of TRPV1 orthologs modulate the threshold for temperature activation. The finding is presented from the perspective of adaptation to cooler environments from amphibians which are poikilothermic vertebrates, hence heavily stressed by environmental temperatures.

By using a comparative approach the authors starts from behavioral experiments and pass through cloning, mutagenesis, and electrophysiological recordings, to finish with molecular modeling and energy calculations to interpret their results.

As far as methods and presentation of the results, the work is well structured. However, the presentation of the problem and the literature review left critical information out. Moreover, the discussion part of the manuscript -although lengthy-, in my opinion, does not offer an adequate perspective and expected insight.

Major Critiques

1) The authors should consider making a better effort on describing the existing literature regarding the role of ankyrin repeats in modulating temperature activation of TRPs. Several works are not mentioned in the introduction and although appear in the discussion section (lines 319-330), they are not used by the authors to provide a suitable molecular interpretation of the results. The question is whether to put together the stepwise hypothesis for activation provided from Lee and Sobolevsky laboratories (refs 20 and 32, the latter only appears in methods), together with the modulatory role of the Ankyrin repeat region as a whole. This would require to put in context the following articles:

10.1073/pnas.1105196108 (ref 12)

10.1073/pnas.1604269113 (ref 14)

10.1016/j.bpj.2019.10.041 (not cited in the text)

10.1113/jp282073 (not cited in the text)

10.1074/jbc.ra120.013037 (cited but out of context, this works refers to pore turret and the authors are using it to support a biochemical work performed in TRPV1's AKN repeats)

The authors will realize that residues from ANK 1 through 6 have been associated to shifts in temperature activation thresholds and some with the energetics associated to channel opening.

2) There is also the problem of interpretation of the results. From Line 37: "Our studies demonstrate that tailed amphibians express highly heat-sensitive TRPV1s". From line 290: "The heat sensitivity of urodelan TRPV1s were all apparently higher compared with the sensitivity of rTRPV1". This reviewer sees this as critical misunderstanding. The sensitivity of an ion channel is observed/analyzed from the slopes of the activation curve. In this case the slope of a current versus temperature curve. The minimal criteria to sustain the used given by the authors would be presenting the Q10s for the different chimeric constructs. Note that "the threshold", is in fact an operational descriptor. Thresholds doesn't have meaning in an energy distribution scheme, we just use the concept as a general descriptor for when the activity surpasses certain open probability. I would recommend to stick to the wording authors are using throughout the manuscript as in reducing or increasing their heat-activation threshold. Otherwise is misleading for the general audience.

3) I would suggest deleting just ANK1 and check whether the channel is just shifting the threshold or losing temperature sensitivity.

4) I would suggest calculating the Q10 for all the different chimeric channels and mutants in order to clarify these issues and draft a more solid discussion.

Minor Critiques

1) Line 46: include TRPC5 as recent works have highlighted its importance in craniofacial pain.

2) Line 56: reference similar works in other thermo-sensitive TRPs. The way the phrase is written gives the idea that this is a unique finding (shifting in temperature thresholds) but in fact was observed in thermoTRPs channels orthologs from elephants, camels, squirrel, birds, fish, etc. A couple are mentioned in line 325. Context is needed at the intro not only at the discussion section.

3) Line 359: The concept "temperature module" is vague.

4) The authors should consider the possibility that the increased flexibility mentioned in line 335, might only contribute to achieve a conformation that is stabilizing the temperature detector elsewhere, as in priming the conformation to detect temperature variations. Following the linear explanation given by the authors, ANK1 would be the temperature detector in TRPV1. To support such statement, bit more than mutagenesis and IV curves should be presented.

Comments of Reviewer #1 and Authors' Responses

Summary: The manuscript by Hori et al. used a comparative approach to discover new aspects of the thermal sensitivity of TRPV1 channels at the molecular level. The author compared the thermal sensitivity of TRPV1 cloned from rats with several amphibian species that prefer cooler environments. These studies revealed that TRPV1 channels cloned from amphibians were much less sensitive to capsaicin activation than rats. Amphibian channels were more sensitive to activation by acidic pH, and the activation temperature for amphibian TRPV1 channels was approximately 10 C lower than that of rat TRPV1 channels. Chimeric and sequence analysis revealed two amino acid residues within the first N-terminal ankyrin repeat domain (ANK1), Q128 and L154, conserved among amphibians but differed from rats. Substitution of these residues into the rat sequence lowered the temperature required for activation and increased sensitivity to capsaicin, and single substitutions had only a small effect. A 3D-modeling approach was used to predict critical residues within ANK1 that contribute to thermal sensitivity. The authors propose that amphibian ANK1 structures containing Q128 and L154 are less stable compared with the same domain in rats, leading to lower temperature activation.

Comments for the authors: This is a fascinating and well-executed study. The data are presented clearly, and my only suggestion for improvement is to reorganize and improve the discussion. In my opinion, this section is somewhat rambling and distracts from an otherwise solid manuscript.

Responses:

We would like to thank the review for careful and critical reading of the manuscript and insightful comments.

- (1) Following the comments by Reviewer #1, the discussion part was largely re-organized and improved to make the flow smooth, for better understanding of the discussion concerning about the roles and function of two residues in ANK1 of TRPV1.
- (2) Description about the relationship between behavioral responses of tailed amphibians to heat and the threshold temperature of TRPV1 was moved to the end (370 on page17 to 386 on page18).
- (3) The discussion on the two residues in ANKs of TRPV1 from squirrel and camel was added before (2) (357 on page16 to 369 on page17).

Comment of Reviewer #2 and Authors' Responses

This manuscript is a follow up on previous works describing a shift in temperature threshold of activation in axolotl TRPV1 channels (Hori, S., & Saitoh, O. BBRC; 2020). In the present work Hori and cols describe an interesting functional-structural observation, that two specific residues located in the first ankyrin repeat of TRPV1 orthologs modulate the threshold for temperature activation. The finding is presented from the perspective of adaptation to cooler environments from amphibians which are poikilothermic vertebrates, hence heavily stressed by environmental temperatures.

By using a comparative approach the authors starts from behavioral experiments and pass through cloning, mutagenesis, and electrophysiological recordings, to finish with molecular modeling and energy calculations to interpret their results.

As far as methods and presentation of the results, the work is well structured. However, the presentation of the problem and the literature review left critical information out. Moreover, the discussion part of the manuscript -although lengthy-, in my opinion, does not offer an adequate perspective and expected insight.

Major Critiques

1) The authors should consider making a better effort on describing the existing literature regarding the role of ankyrin repeats in modulating temperature activation of TRPs. Several works are not mentioned in the introduction and although appear in the discussion section (lines 319-330), they are not used by the authors to provide a suitable molecular interpretation of the results. The question is whether to put together the stepwise hypothesis for activation provided from Lee and Sobolevsky laboratories (refs 20 and 32, the latter only appears in methods), together with the modulatory role of the Ankyrin repeat region as a whole. This would require to put in context the following articles:

10.1073/pnas.1105196108 (ref 12)

10.1073/pnas.1604269113 (ref 14)

10.1016/j.bpj.2019.10.041 (not cited in the text)

10.1113/jp282073 (not cited in the text)

10.1074/jbc.ra120.013037 (cited but out of context, this works refers to pore turret and the authors are using it to support a biochemical work performed in TRPV1's AKN repeats)

The authors will realize that residues from ANK 1 through 6 have been associated to shifts in temperature activation thresholds and some with the energetics associated to channel opening.

Responses:

We would like to thank Reviewer #2 for careful reading of the manuscript and insightful criticisms. Following the comments, we now cited and described the findings of previous works, including 5 references reviewer#2 pointed out (ref 9,10,12,13,14), to introduce and the present molecular interpretation of heat-activation mechanism of TRPV1 (27 to 46 on page3).

10.1073/pnas.1105196108 (ref 12) ---- ref 9 in the revised introduction

10.1073/pnas.1604269113 ---- ref13 in the revised introduction

10.1016/j.bpj.2019.10.041 ---- ref14 in the revised introduction

10.1113/jp282073 ---- ref10 in the revised introduction

10.1074/jbc.ra120.013037 ---- ref12 in the revised introduction

As to the structural determinant(s) of the thermal activation, the contributions of distinct regions of TRPV1 have been reported as follows. (1) The deletions of C-terminal of TRPV1 lowered the threshold for the thermal activation^{6, 7,8}. (2) The membrane-proximal domain (MPD) connecting N-terminal Ankyrin repeats (ANKs) and the first transmembrane domain determines the temperature dependence of channels^{9, 10}. (3) Heat activation and shifts of threshold temperature is intrinsic in

the pore domain of TRPV1^{11, 12}. (4)The ANKs domain is important for heat activation of TRPV1^{13, 14}. From these functional studies, it appears that molecular determinants of hypothetical heat-dependent activation may rather spread over the TRPV1 molecule.

More recently, Nadezhdin et al. reported the cryo-EM structures of heat-activated TRPV3¹⁵, and Kwon et al. showed the cryo-EM structures of heat-activated TRPV1¹⁶. Although detailed information of conformational changes are different, both studies demonstrated that the mechanism of heat activation includes two steps, and that global conformational changes across multiple topologically-distant subdomains of TRP channel might be followed by the rearrangement of the pore to lead to gate opening. Therefore, since each distinct domains might be involved after heat stimulation, it is thought that mutations that affect heat-sensing or coupling mechanisms could not be functionally distinguishable. Thus, the primary heat-sensing module(s) which determine the threshold temperature of TRPV1 still remains unclear at this stage.

2) There is also the problem of interpretation of the results. From Line 37: “Our studies demonstrate that tailed amphibians express highly heat-sensitive TRPV1s”. From line 290: “The heat sensitivity of urodelan TRPV1s were all apparently higher compared with the sensitivity of rTRPV1”. This reviewer sees this as critical misunderstanding. The sensitivity of an ion channel is observed/analyzed from the slopes of the activation curve. In this case the slope of a current versus temperature curve. The minimal criteria to sustain the used given by the authors would be presenting the Q10s for the different chimeric constructs. Note that “the threshold”, is in fact an operational descriptor. Thresholds doesn’t have meaning in an energy distribution scheme, we just use the concept as a general descriptor for when the activity surpasses certain open probability. I would recommend to stick to the wording authors are using throughout the manuscript as in reducing or increasing their heat-activation threshold. Otherwise is misleading for the general audience.

Responses:

We would like to thank Reviewer #2 for pointing out this important point. In the revised manuscript, we avoided using the term of “the heat sensitivity”, and consistently used e.g. “reduced (or increased) the temperature threshold for heat-activation”. These changes were indicated by red characters in the revised manuscript.

3) I would suggest deleting just ANK1 and check whether the channel is just shifting the threshold or losing temperature sensitivity.

Responses:

Following the comment of Reviewer #2, we constructed two deletion mutants of axolotl TRPV1 (deletion of N-terminal part and ANK1 (1-166 aa, Δ N), and deletion of just ANK1 (127-166aa, Δ ANK1)). Since capsaicin, acid, or heat could not at all activate these mutants, we judged they are non-functional. We also made two deletion mutants of rat TRPV1 (deletion of N-terminal part and ANK1 (1-152aa, Δ N), and deletion of just ANK1 (113-152aa, Δ ANK1)). These deletion mutants of rat TRPV1 were also non-functional. These data are shown in [Figure only for reviewers]. As a result, we could not examine the effect of the deletion of ANK1 on the temperature threshold.

4) I would suggest calculating the Q10 for all the different chimeric channels and mutants in order to clarify these issues and draft a more solid discussion.

Responses:

We agree with Reviewer#2 that the Q10 value well reflects the temperature sensitivity. However, we think the accurate determination of Q10 of TRPV1 channel is not practically straightforward by the following reasons.

(1) For the solid and trustable calculation of Q10 of TRPV1 channel, it is necessary to isolate pure TRPV1 channel current. If the expression level of TRPV1 channel is not high enough and the background current from other (temperature insensitive) channels are significantly included, the change of the total current amplitude apparently looks less steep and Q10 looks smaller.

(2) Also, Q10 is determined from the slope of the linear fit of the log plot of the current amplitude in the temperature range above the threshold, and the value strongly depends on the accuracy of the fitting. Just a slight difference of the fitting results in large change of Q10.

We actually calculated the Q10 values of WT and various mutants from the obtained data, but the data showed very high variations. We judged the calculated values are not necessarily trustable by the reasons above, and that the inclusion might induce confusion. Thus, we decided not to present the Q10 values data. As already written in our response to the major comment #2, we strictly avoided using “temperature sensitivity” as we do not show Q10, and consistently used “change of the temperature threshold of activation”. We understand that the temperature threshold is not a physical parameter, but it is a good indicator to describe our findings.

Minor Critiques

1) Line 46: include TRPC5 as recent works have highlighted its importance in craniofacial pain.

Responses:

As suggested, TRPC5 was included (23 on page2).

2) Line 56: reference similar works in other thermo-sensitive TRPs. The way the phrase is written gives the idea that this is a unique finding (shifting in temperature thresholds) but in fact was observed in thermoTRPs channels orthologs from elephants, camels, squirrel, birds, fish, etc. A couple are mentioned in line 325. Context is needed at the intro not only at the discussion section.

Responses:

As suggested, the sentence “The control mechanism for the threshold temperature of TRPV1, however, is not well understood.” was removed. The following descriptions were included in the introduction (50-58 on page4).

Investigation of TRPV1 from ground squirrels and camels, which can tolerate higher environmental temperatures, revealed that these animals express TRPV1 with a dramatically decreased thermosensitivity¹³. Furthermore, characterization of a cold-sensitive thermoTRP, TRPM8 from elephants and penguins indicated that penguin’s TRPM8 exhibited much decreased cold-activated currents¹⁷. In these reports, several residues and structures involved in tuning thermal activation in thermoTRPs have been reported, such as Asn126 and Glu190 of

squirrel TRPV1 in the N-terminus, Asn124 and Glu188 of camel TRPV1 in the N-terminus¹³, and Y919 of penguin TRPM8 in the pore domain¹⁷.

3) Line 359: The concept “temperature module” is vague.

Responses:

As suggested, “temperature module” was changed to “a structural module which contributes to the control of the temperature sensitivity”(320-321 on page15).

4) The authors should consider the possibility that the increased flexibility mentioned in line 335, might only contribute to achieve a conformation that is stabilizing the temperature detector elsewhere, as in priming the conformation to detect temperature variations. Following the linear explanation given by the authors, ANK1 would be the temperature detector in TRPV1. To support such statement, bit more than mutagenesis and IV curves should be presented.

Responses:

We would like to deeply thank Reviewer#2 for the valuable comment and constructive suggestion of additional experiments.

We performed new biochemical experiments to examine if the thermal structure stability of the N-terminus of TRPV1 can be controlled by the point mutations at the 2nd and 28th position of ANK1 (Fig. 10). Results clearly demonstrated that mutation on the ANK1 which drastically shifts the thermal threshold of TRPV1 apparently changes the heat-dependent increase in flexibility of the purified N-terminus protein, strongly supporting our proposed mechanism based on the electrophysiological data. These results are included in the RESULTS section (2.6, 268-286 on page13) and discussed in the DISCUSSION section (318-328 on page15). The related methodology is also added in the method section (500 on page22 to 521 on page23). Thanks to the valuable comment by Reviewer#2, we believe the scientific merit of the work has been significantly elevated by the addition of this new data.

2.6. The mutation identified by the electrophysiological experiments also changes the thermal stability of the purified N-terminus protein containing ANKs of TRPV1

Recently, using biochemically purified N-terminus protein containing whole ANKs from TRPV1 (100-362aa) in solution, the thermal shift unfolding with SYPRO orange, circular dichroism, and tryptophan fluorescence measurements were performed and it was revealed that the N-terminus undergoes apparent structure changes in accordance with an increase in the temperature that may lead to the TRPV1 channel activation¹⁴. Our electrophysiological data demonstrated that R114Q and K140L mutations in ANK1 in the N-terminus of rTRPV1 significantly lowered changed the heat activation threshold (Fig. 6a, b). Therefore, we examined whether R114Q and K140L mutations really affect the thermal stability of the purified rTRPV1 N-terminus protein by the thermal shift unfolding (Fig. 10). The melting temperature, T_m for the wild type N-terminus was $32.03 \pm 0.01^\circ\text{C}$, which is close to the reported value¹⁴. Two mutations significantly decreased T_m values of the thermal shift unfolding of the N-terminus (R114Q: $30.16 \pm 0.02^\circ\text{C}$, K140L: $28.04 \pm 0.06^\circ\text{C}$), indicating that the thermal stability was down-

regulated by the mutation (Fig. 10). These results demonstrate that the protein structure stability of the N-terminus is efficiently controlled by two amino acid residues at the 2nd and 28th position of ANK1 of TRPV1, and strongly suggest that the ANK1 functions as a structural module which determines the heat-activation threshold of TRPV1.

Previously, using the biochemically isolated ANK domain of rTRPV1, the temperature-dependent dynamics of protein conformation were observed¹⁴, and a possibility that the ANK domain of TRPV1 may function as a structural module which controls the temperature sensitivity was demonstrated. Therefore, we prepared the purified protein of the N-terminus containing the ANKs of TRPV1 wild-type and mutants, and analyzed the conformation, thermal stability, and effects of mutations on protein stability and heat-dependency. Results clearly demonstrated that the protein structure stability of the N-terminus is efficiently controlled by two residues at the 2nd and 28th position of ANK1 of TRPV1, and strongly suggests that the ANK1 functions as a structural module determining the heat-activation threshold of TRPV1.

Reviewers' Comments:

Reviewer #2:

Remarks to the Author:

The authors have done a great job with this revision, and my concerns have been fully addressed. Congratulations on a great study.

Reviewer #3:

Remarks to the Author:

In the new version, the authors addressed all my concerns.